# Whole Exome Sequencing Points towards a Multi-Gene Synergistic Action in the Pathogenesis of Congenital Combined Pituitary Hormone Deficiency

**DOI:** 10.3390/cells11132088

**Published:** 2022-06-30

**Authors:** Amalia Sertedaki, Elizabeth Barbara Tatsi, Ioannis Anargyros Vasilakis, Irene Fylaktou, Eirini Nikaina, Nicoletta Iacovidou, Tania Siahanidou, Christina Kanaka-Gantenbein

**Affiliations:** 1Division of Endocrinology, Diabetes and Metabolism, Center for Rare Paediatric Endocrine Diseases, First Department of Pediatrics, Medical School, “Aghia Sophia” Children’s Hospital, National and Kapodistrian University of Athens, 11527 Athens, Greece; etatsi@med.uoa.gr (E.B.T.); vasilakisioan@yahoo.gr (I.A.V.); efylaktou@med.uoa.gr (I.F.); ckanaka@med.uoa.gr (C.K.-G.); 2Neonatology Unit, First Department of Pediatrics, Medical School, “Aghia Sophia” Children’s Hospital, National and Kapodistrian University of Athens, 11527 Athens, Greece; enikaina@gmail.com (E.N.); siahan@med.uoa.gr (T.S.); 3Department of Neonatology, Medical School, Aretaieion Hospital, National and Kapodistrian University of Athens, 11528 Athens, Greece; niakobid@med.uoa.gr

**Keywords:** combined pituitary hormone deficiency, congenital hypopituitarism, hypogonadotropic hypogonadism, whole exome sequencing, pathogenic gene variants

## Abstract

Combined pituitary hormone deficiency (CPHD) is characterized by deficiency of growth hormone and at least one other pituitary hormone. Pathogenic variants in more than 30 genes expressed during the development of the head, hypothalamus, and/or pituitary have been identified so far to cause genetic forms of CPHD. However, the etiology of around 85% of the cases remains unknown. The aim of this study was to unveil the genetic etiology of CPHD due to congenital hypopituitarism employing whole exome sequencing (WES) in two newborn patients, initially tested and found to be negative for *PROP1*, *LHX3*, *LHX4* and *HESX1* pathogenic variants by Sanger sequencing and for copy number variations by MLPA. In this study, the application of WES in these CPHD newborns revealed the presence of three different heterozygous gene variants in each patient. Specifically in patient 1, the variants *BMP4*; p.Ala42Pro, *GNRH1*; p.Arg73Ter and *SRA1*; p.Gln32Glu, and in patient 2, the *SOX9*; p.Val95Ile, *HS6ST1*; p.Arg306Gln, and *IL17RD*; p.Pro566Ser were identified as candidate gene variants. These findings further support the hypothesis that CPHD constitutes an oligogenic rather than a monogenic disease and that there is a genetic overlap between CPHD and congenital hypogonadotropic hypogonadism.

## 1. Introduction

Combined pituitary hormone deficiency (CPHD) is characterized by deficiencies of growth hormone (GH) and at least one more anterior pituitary hormone. The most frequent hormone deficiencies besides that of GH are deficiencies of thyroid stimulating hormone (TSH), the gonadotrophins, luteinizing hormone (LH), follicle stimulating hormone (FSH), and lastly, adrenocorticotropic hormone (ACTH). The normal secretion of LH and FSH is under the control of the gonadotropin releasing hormone (GnRH) produced by the hypothalamus that binds to GnRH receptors in the anterior pituitary. Reduced production and secretion of gonadotrophins leads to hypogonadotropic hypogonadism (HH) [1]. CPHD may either present already in the neonatal period, representing cases of congenital hypopituitarism, or later during life secondary to insults of the hypothalamic–pituitary region, such as tumors or irradiation, leading to the acquired forms of CPHD.

Cases of congenital hypopituitarism are, therefore, diagnosed during the neonatal period, most commonly with the clinical picture of hypoglycemia, due to GH and ACTH/cortisol deficiency, protracted jaundice, due to TSH/thyroxine and cortisol deficiency and, especially in males, micropenis, due to GH and gonadotrophin deficiencies.

In less severe cases, if left undiagnosed beyond the neonatal period, the patient with congenital hypopituitarism may be diagnosed later on with short stature as the first clinical manifestation accompanied by other features such as delayed puberty or metabolic abnormalities.

Neuroimaging may reveal abnormalities of the hypothalamic–pituitary region, mainly hypoplasia of the anterior pituitary, ectopic posterior pituitary, and hypoplastic or absent pituitary stalk resulting in impaired pituitary hormone secretion, while some cases are further characterized by craniofacial malformations [2]. Thus, depending on the hormone deficiencies identified and the underlying etiology, there is a wide spectrum of phenotypic characteristics ranging from mild to more severe. The incidence of congenital forms of CPHD is estimated at 1:8000 live births worldwide [3].

As stated above, the etiology of CPHD might be either genetic or acquired due to trauma, tumor, infection, infiltration or autoimmune diseases affecting the hypothalamic–pituitary region [3,4,5]. As far as the genetic etiology is concerned, several genes, mostly transcription factors, have been identified to be associated with congenital forms of CPHD and participate in the Bmp, Shh, Fgf, Wnt or Notch signaling pathways [6]. Pathogenic variants of the transcription factor genes *PROP1*, *POU1F1*, *HESX1*, *LHX3*, and *LHX4* are the most frequent causes of CPHD [7].

In recent years, the application of next generation sequencing methodologies (NGS), including the screening of large gene panels in many patients simultaneously, and whole exome sequencing have revealed new variants of known CPHD genes and new genes associated with CPHD [8,9,10]. However, the etiology of approximately 85% of the cases still remains unknown [3]. Furthermore, various studies demonstrated that the genetic causes of CPHD and pituitary stalk interruption syndrome (PSIS) are oligogenic rather than monogenic [11,12,13]. Similarly, congenital hypogonadotropic hypogonadism (CHH) is now considered as an oligogenic disease, and it has been shown by various studies that mutations in genes associated with CHH may also be related with CPHD, PSIS and holoprosencephaly, indicating a genetic overlap between these syndromes [14,15,16].

The aim of this study was to unveil the genetic etiology in two newborn patients with congenital hypopituitarism, employing WES.

## 2. Materials and Methods

### 2.1. Patients

Two male (46, XY) newborns, diagnosed with CPHD, were referred to the Laboratory of Molecular Endocrinology, Division of Endocrinology, Diabetes and Metabolism, Center for Rare Pediatric Endocrine Diseases, First Department of Pediatrics, National and Kapodistrian University of Athens, School of Medicine, “Aghia Sophia” Children’s Hospital, for genetic testing. The parents of the patients provided written informed consent.

Patient 1 was delivered by caesarean section due to intrauterine growth restriction (IUGR) with a birth weight of 2200 g. Shortly after birth, he presented with refractory hypoglycemia and mild hypotonia. On physical examination, he had micropenis (1 cm) with bilaterally palpable small testes (testes volume: 1 mL measured by orchidometer). Endocrinological workup revealed secondary hypothyroidism, secondary adrenal insufficiency, and hypogonadotropic hypogonadism (HH) (Table 1).

MRI scan of the hypothalamic–pituitary region depicted hypoplastic anterior pituitary and ectopic posterior pituitary with absence of the pituitary stalk. The patient was put on substitutive treatment with hydrocortisone, testosterone during the mini-puberty period, levothyroxine and somatropin.

Patient 2 presented with protracted jaundice, direct hyperbilirubinemia, severe episodes of hypoglycemia and severe hypertransaminasemia. Clinical examination revealed a hypoplastic and thin penis (1.5 cm). Ultrasound of the scrotum depicted hypoplastic testicles bilaterally with hydrocele on the right. Hormonal assessment revealed multiple pituitary hormone deficiencies with central hypothyroidism, secondary adrenal insufficiency, hypogonadotropic hypogonadism, and growth hormone deficiency (Table 1). MRI scan depicted a hypoplastic anterior pituitary, an ectopic posterior pituitary lobe and thinning of the pituitary stalk. The patient was treated with substitution of hydrocortisone, thyroxine, testosterone during the mini-puberty period, and somatropin.

Both patients were tested by direct sequencing of the *PROP1*, *LHX3*, *LHX4* and *HESX1* genes, with no pathogenic variant identified. They were also found to be negative for copy number variants (CNV) of the genes *GH1*, *POU1F1*, *GHRHR*, *LHX3*, *LHX4*, *HESX1* and *PROP1* employing multiplex ligation-dependent probe amplification (MLPA) (SALSA MLPA Probemix P216 Growth Hormone Deficiency mix-1; MRC-Holland, Amsterdam, The Netherlands).

The study was approved by the Institutional Scientific and Bioethics Committee and is in accordance with the Declaration of Helsinki.

### 2.2. DNA Isolation

DNA was extracted from 400 μL peripheral blood leucocytes employing the Maxwell 16 Blood DNA Purification Kit (Promega, Madison, WI, USA), an automated method that uses paramagnetic particles to bind DNA, according to the manufacturer’s instructions.

### 2.3. Whole Exome Sequencing, Filtering and Bioinformatic Analysis

WES was carried out on an Ion S5™ XL System (ThermoFisher Scientific, Waltham, MA, USA) using the Ion AmpliSeq™ Exome RDY Kit (ThermoFisher Scientific, Waltham, MA, USA). The amount of DNA used for WES was ≥50 ng.

Base calling, alignment to the human genome assembly hg19 and variant calling were performed on Torrent Suite^TM^ Server according to the manufacturer’s instructions. The variants were annotated by: (1) Ion Reporter (v5.2.0.66; https://ionreporter.thermofisher.com, accessed 17 May 2019) and (2) ANNOVAR (http://annovar.openbioinformatics.org accessed 17 May 2019) through VarAFT software (Variant Annotation and Filter Tool, Version 2.15; https://varaft.eu/, accessed 16 September 2019 and 9 February 2021).

Initial filtering was carried out employing an in silico panel of 194 genes related to CPHD and isolated growth hormone deficiency (IGHD): *ARNT2*, *BMP2*, *BMP4*, *CDON*, *CHD7*, *FGF10*, *FGF18*, *FGF8*, *FGFR1*, *GATA2*, *GHRH*, *GHRHR*, *GLI2*, *GLI3*, *GLI4*, *GPR161*, *HESX1*, *HHIP*, *HNRNPU*, *IFT172*, *IGSF1*, *LHX3*, *LHX4*, *NEUROD1*, *NFKB2*, *NR5A1*, *OTX2*, *PAX6*, *PCSK1*, *PITX1*, *PITX2*, *PNPLA6*, *POLR3A*, *POU1F1*, *PROK1*, *PROKR2*, *PROP1*, *RBM28*, *ROBO1*, *SHH*, *SIX1*, *SIX2*, *SIX3*, *SIX4*, *SIX5*, *SIX6*, *SOX1*, *SOX2*, *SOX3*, *TBX19*, *TCF7L1*, *TGIF*, *TGIF1*, *TRH*, *TRHR*, *TSHB*, *WDR11*, *WNT4*, *WNT5A*, *ZFHX3*, *ZSWIM6*; HH: *ANOS1*, *AXL*, *CCDC141*, *DCAF17*, *DMXL2*, *DUSP6*, *FEZF1*, *FGF17*, *FLRT3*, *GNRH1*, *GNRHR*, *HS6ST1*, *IL17RD*, *KAL1*, *KISS1*, *LEPR*, *LEP*, *LHB*, *NR0B1*, *NSMF*, *PROK2*, *SEMA3A*, *SOX10*, *TAC3*, *TACR3*, *OTUD4*, *RNF216*, *SEMA3E*, *SEMA7A*, *SRA1*; as well as holoprosencephaly (HPE) and Corpus Callosum Agenesis: *AHI1*, *AKT3*, *AMPD2*, *ANOP1*, *ARID1B*, *ARL13B*, *ARX*, *ASPM*, *ATR*, *ATRX*, *B9D2*, *BCOR*, *CASK*, *CC2D2A*, *CENPJ*, *CEP152*, *CEP290*, *CEP41*, *CEP63*, *CREBBP*, *CTBP1*, *DCX*, *DHCR24*, *DHCR7*, *DIS3L2*, *DISC1*, *DISP1*, *DLL1*, *EFNB1*, *EOMES*, *EP300*, *EPG5*, *FGFR2*, *FH*, *FKRP*, *FKTN*, *FLNA*, *FOXH1*, *GAS1*, *GPSM2*, *GTDC2*, *HCCS*, *HHAT*, *HYLS1*, *IGBP1*, *IGF1*, *INPP5E*, *ISPD*, *KAT6B*, *KCC3*, *KIF7*, *L1CAM*, *LRP2*, *MED1*, *MED12*, *MID1*, *MKS1*, *NDE1*, *NFIX*, *NIN*, *NODAL*, *NPHP1*, *NPHP3*, *NSD1*, *OFD1*, *OTX1*, *PDHA1*, *PDHB*, *POMGNT1*, *POMT1*, *POMT2*, *PTCH*, *PYCR1*, *RAB18*, *RAB3GAP1*, *RAB3GAP2*, *RBBP8*, *RBM10*, *RELN*, *RNU4ATAC*, *RPGRIP1L*, *RPS6KA3*, *SCKL3*, *SLC12A6*, *SPG11*, *SPRY4*, *STRA6*, *SUFU*, *TCTN1*, *TCTN2*, *TCTN3*, *TMEM138*, *TMEM216*, *TMEM237*, *TMEM67*, *TUBA1A*, *TUBB2B*, *TUBB3*, *VAX1*, *ZIC2*, to select variants with minor allele frequency (MAF) values < 2% and depth ≥ 20× (filter 1). Next, variants resulting in premature stop codons with depth ≥ 20× (filter 2) as well as homozygous and heterozygous variants in coding regions and splice sites with MAF values < 1% and depth ≥ 30× (filter 3) were also checked (Figure 1).

Another 233 genes found after relevant literature search were checked for pathogenic variants: *ACVRL1*, *AES*, *AGAP1*, *AHR*, *AKAP2*, *AKT1S1*, *ALK*, *ALMS1*, *AMN1*, *ANK1*, *ANOS1*, *ARNT2*, *ASCL1*, *ATF2*, *ATP13A1*, *AXIN1*, *BAMBI*, *BMP1*, *BMPER*, *BMPR1A*, *BMPR1B*, *BOD1L1*, *CARTPT*, *CCKBR*, *CCN4*, *CDY1*, *CEBPB*, *CGA*, *CHRD*, *CPE*, *CRIM1*, *CRY1*, *CSNK1A1*, *CSNK1D*, *CSNK2A1*, *CTNNB1*, *CXCR4*, *DAAM1*, *DEK*, *DHH*, *DISP3*, *DKK3*, *DLK1*, *DLL3*, *DNER*, *DR1*, *DVL1*, *DVL2*, *E2F4*, *EID1*, *ENG*, *EYA4*, *EZH2*, *FAAH2*, *FANCB*, *FANCC*, *FANCD2*, *FBXW2*, *FGF1*, *FGF13*, *FGF14*, *FGF4*, *FGFR1OP*, *FGFR1OP2*, *FIBP*, *FKBP8*, *FRAT2*, *FRS3*, *FSTL1*, *FSTL5*, *FUS*, *FZD2*, *FZD4*, *FZD5*, *FZD6*, *GAP43*, *GDNF*, *GH*, *GH1*, *GLG1*, *GLI1*, *GLIPR2*, *GRK1*, *HES6*, *HEY1*, *HIF1A*, *HMGA2*, *HMX2*, *HMX3*, *ID1*, *ID2*, *ID3*, *ID4*, *IGSF10*, *IL6ST*, *IRS4*, *ISL1*, *ITGA2B*, *JAG1*, *JUN*, *KISS1R*, *KLF4*, *LAMB2*, *LDB1*, *LHX1*, *LHX2*, *LHX8*, *LTBP3*, *LTBP4*, *MAMLD1*, *MASTL*, *MECP2*, *MITOL*, *MLH3*, *MYO9B*, *NAALADL1*, *NBL1*, *NEUROG2*, *NKX2-1*, *NLE1*, *NOBOX*, *NOS1*, *NOTCH1*, *NOTCH2*, *NPC1*, *NPFFR1*, *NPVF*, *NR4A1*, *NRP2*, *OTP*, *OVOL1*, *PALLD*, *PAX2*, *PDE3A*, *PIAS4*, *PIK3CA*, *PJA1*, *PLEKHA5*, *PLPP3*, *PLXND1*, *POFUT1*, *POLR3B*, *POLR3D*, *POU3F2*, *PPAP2B*, *PPHLN1*, *PPM1L*, *PPRC1*, *PRKAG1*, *PRKAR1A*, *PTCH1*, *PTCHD2*, *PTTG1*, *RARA*, *RB1*, *RBPJ*, *RD3*, *RNPC3*, *ROR2*, *RORA*, *RXRA*, *SENP2*, *SFRP1*, *SIM1*, *SIRT1*, *SLC15A4*, *SLC20A1*, *SLC9A3R1*, *SLX4*, *SMAD1*, *SMAD2*, *SMAD3*, *SMAD4*, *SMAD6*, *SMAD7*, *SMARCA4*, *SMURF1*, *SNIP1*, *SOSTDC1*, *SP1*, *SP3*, *SPIN2A*, *SPRED2*, *SPRY1*, *SPRY2*, *SRF*, *SRY*, *STAT3*, *STAT5B*, *STIL*, *STRAP*, *STUB1*, *SYNPO2*, *TAB1*, *TANC2*, *TBX3*, *TCF12*, *TCF3*, *TCF4*, *TDGF1*, *TET2*, *TGFB2*, *TGFBI*, *TGFBR1*, *TGFBR2*, *TLE4*, *TLE5*, *TMEM231*, *TNRC6A*, *TP53*, *TRAPPC9*, *TRIP11*, *TSHZ1*, *TSPAN11*, *TSPYL2*, *TWSG1*, *TXNDC5*, *WDR12*, *WIF1*, *WISP1*, *WNT5B*, *WNT6*, *WNT9A*, *YBX1*, *ZBTB20*, *ZEB1*, *ZFY*, *ZFYVE9*, *ZIC1*, *ZNF488*, *ZNF8*, *ZSWIM6*.

Ion Reporter and VarAFT software gathered the pathogenicity scores from 15 predictive bioinformatic tools. The following in silico tools were further employed for the interpretation of the filtered variants: protein analysis through evolutionary relationships (Panther; http://www.pantherdb.org), and Human Splicing Finder (HSF; http://www.umd.be/HSF3/). The variants were also searched in the VarSome (https://varsome.com/), Ensembl (http://www.ensembl.org), Human Gene Mutation Database (HGMD; www.hgmd.cf.ac.uk) and Single Nucleotide Polymorphism Database (dbSNP; www.ncbi.nlm.nih.gov/snp). These in silico tools were accessed various times over a 2 years period (2019–2021). The classification of the variants’ pathogenicity was performed taking into consideration the information gathered from the above-mentioned databases and in silico tools for the application of the American College of Medical Genetics (ACMG) criteria. The nucleotide sequences were visualized using the software IGV 2.3.91 (Broad Institute; https://software.broadinstitute.org/software/igv/, September 2019 and February 2021).

To verify candidate variants, Sanger sequencing was carried out for the patients and their parents. All variants reported herein were confirmed by Sanger sequencing the corresponding exon of each gene in the patients and their parents.

The frequencies of the candidate variants identified in our patients were also searched in a 500 Greek WES in-house dataset.

## 3. Results

### 3.1. Patient 1

Employing WES, 18,921 genes were sequenced with 90.72% coverage (>20×), and 38,348 variants in 12,459 genes were detected.

Three heterozygous variants in the genes, *BMP4*, *GNRH1* and *SRA1* related to patient’s phenotype, were identified (Table 2).

Notes: CADD: value > 20 = Deleterious, GERP: −12.3 = less conserved to 6.17 = more conserved, ACMG classification: P = Pathogenic, LP = Likely Pathogenic, and VUS = Variant of Uncertain Significance. The *BMP4* variant c.124G > C, p.Ala42Pro (193 ×), of maternal inheritance, was identified by the CPHD gene panel employed. Residue Ala42 is conserved and the p.Ala42Pro is a rare variant (f = 0.000283). According to ACMG criteria, it is classified as Variant of Unknown Significance (VUS).

The *GNRH1* variant c.217C > T, p.Arg73Ter (58×), of paternal inheritance, was identified by the CHH gene panel. It is a nonsense variant resulting in protein truncation by 19 amino acids. It is an ultrarare variant with frequency in gnomAD Exomes 0.00000805. According to ACMG criteria, it is classified as likely pathogenic.

The *SRA1* variant c.94C > G, p.Gln32Glu (200×), of maternal inheritance, was identified by the CHH gene panel. Residue Gln32 is highly conserved among species, and the in silico conservation tools Panther and GERP classified this variant as pathogenic. According to ACMG criteria, it is classified as VUS.

### 3.2. Patient 2

Employing WES, sequencing was achieved for 18,900 genes with 88.92% coverage (>20×), and 38,032 variants in 12,337 genes were detected.

Three heterozygous variants in the genes *SOX9*, *HS6ST1* and *IL17RD*, related to patient’s phenotype, were identified (Table 2).

The novel *SOX9* gene variant c.283G > A, p.Val95Ile (82×), of maternal inheritance, was identified by the CHH gene panel. It was found to be pathogenic by PolyPhen2, Mutation Taster, DANN and CADD bioinformatic tools. Residue Val95 is conserved, and the in silico conservation tools Panther and GERP classified it as pathogenic. According to ACMG criteria, it is classified as VUS.

The *HS6ST1* gene variant c.917G > A, p.Arg306Gln (143×), of maternal inheritance, was identified by the CHH gene panel employed. According to the ACMG criteria, it is classified as pathogenic.

The *IL17RD* gene variant c.1696C > T, p.Pro566Ser (37×), of maternal inheritance, was identified by the CHH gene panel. Its gnomAD Exomes frequency is 0.0144, and it is classified as VUS according to ACMG criteria.

## 4. Discussion

The genetic etiology of CPHD is heterogeneous, and patients present complex phenotypes. Despite the wide application of WES in large cohorts of patients, most cases still remain without genetic identification. In recent years, various studies revealed a possible oligogenic or multigenic basis for CPHD.

In this study, the application of WES in two CPHD newborns revealed the presence of several gene variants mostly associated with CHH and the identification of a novel SOX9 variant.

Patient 1 was found to be heterozygote for variants in three genes, the *BMP4*, *GNRH1* and *SRA1*.

The bone morphogenic protein-4 (BMP4), a member of the BMP family that belongs to the transforming growth factor-beta (TGF-β) family, plays a significant role in early organogenesis, pituitary development and function [17]. BMP4 activity is required for initial organ commitment and development of the pituitary anlage. Particularly BMP4 expression in early development contributes to the formation of Rathke’s pouch in the mouse, while BMP4 together with BMP2, and 7 later secreted by surrounding tissues, contribute to the polarization of the pouch [18]. Inhibition of Bmp signaling in the oral ectoderm of mouse embryos disrupts pituitary cell specification [19].

The *BMP4* gene encodes a 408 amino acid protein that consists of an *N*-terminal signal peptide, which directs the protein to the secretory pathway, a prodomain that ensures proper folding, and the C-terminal mature peptide [20].

The p.Ala42Pro *BMP4* gene variant identified in our patient is located in the highly conserved prodomain part of the protein and up to now has been classified as a VUS according to the ACMG criteria. Various *BMP4* variants have been reported in patients with tooth agenesis, amongst them the *BMP4* variant p.Ala42Pro identified in our patient and a female patient and her mother with tooth agenesis [21,22]. Other variants also found in the prodomain region have been reported in patients with a wide spectrum of phenotypes, such as Stickler syndrome, anophthalmia, ocular malformation, cleft palate and scleroma, as well as colorectal cancer predisposition [23,24,25,26].

*BMP4* gene variants have been associated with a wide range of phenotypic characteristics, including ocular, craniofacial, finger and urogenital abnormalities, and psychomotor retardation, as well as CPHD [23,27,28,29,30]. Specifically, the *BMP4* variants p.Arg300Pro (VUS) and de novo p.Trp265Ter have been identified in heterozygosity in two patients with CPHD; on neuroimaging, these patients presented hypoplastic anterior pituitary gland, ectopic posterior pituitary, and absent pituitary stalk, as is the case in our patient, while clinically they presented sclerosed nodules at the hands, short metacarpals IV, azoospermia [29] and, GH and TSH deficiencies [30]. After exome sequencing in 20 patients (trio analysis) with isolated pituitary stalk interruption syndrome (PSIS), the *BMP4* variant p.Ala334Asp was identified together with a *SIX6* gene variant p.Glu129Lys (and *DCHS1* gene variants) in a patient who presented with growth retardation, pituitary hormone deficiencies and MRI findings of ectopic posterior pituitary, small stalk and anterior pituitary [11].

The *GNRH1* gene encodes a preproprotein that generates the GnRH decapeptide, a member of the gonadotropin-releasing hormone (GnRH) family of peptides. It consists of four exons, three of which are translated, and encodes a 92 amino acid preprohormone that is organized to a signal sequence, followed by the GnRH decapeptide, a conserved GKR cleavage site, and the GnRH-associated peptide (GAP). The decapeptide is secreted in a pulsatile manner by the GnRH neurons in the hypothalamus and triggers the synthesis and secretion of LH and FSH by the anterior pituitary gonadotrophs [31].

The *GNRH1* pathogenic variant p.Arg73Ter was identified in heterozygosity in our patient 1, inherited from his father. Residue R73 resides in the GAP region of the protein and results in its truncation by 19 amino acids, and although a nonsense variant, it is considered as VUS since the resulting protein truncation leaves the GnRH decapeptide intact.

This variant has been previously reported in heterozygosity in a male patient investigated for absent puberty at the age of 17 ^6/12^ years [32]. Chan and coworkers speculated that, since the premature stop codon is close to a splice junction, it may escape nonsense-mediated decay and, if this is the case, an abnormal protein is produced. Although, the effect of this variant on GNRH1 synthesis is unclear, the authors suggested that the CHH of this family with the p.Arg73Ter might be of autosomal dominant inheritance with variable expressivity since the father, heterozygous for the variant, exhibited a mild phenotype of delayed puberty [32].

Furthermore, recently a homozygous splice site variant of exon 2 (c.142-2A > C), damaging the natural acceptor site and activating a cryptic acceptor site, was described in a patient with lack of pubertal development and bilateral cryptorchidism, surgically corrected in childhood. In the patient’s lymphocytes, an aberrant *GNRH1* transcript was detected, harboring a four bp deletion resulting in a premature stop codon at amino acid 74. The authors speculated that this may lead to mRNA nonsense mediated decay indicating a probable functional importance of the GAP region of *GNRH1* [33].

Only a few *GNRH1* gene pathogenic variants have been detected to date in patients with CHH, even though GnRH is the central regulator in reproductive function and sexual development. Thus far, four variants have been identified in the decapeptide encoding sequence, the p.Gly29GlyfsTer12 and p.Arg31His in homozygosity, and the p.Trp26Ter and p.Arg31Cys in heterozygosity [32,34,35]. The remaining variants have been found in heterozygosity either in the highly conserved signal peptide (p.Val18Met) or in the GAP (p.Thr58Ser and p.Arg73Ter) [32].

Our patient was very young (infant), and his severe micropenis and hypoplastic testicles along with the absence of LH and FSH surge during his mini-puberty suggest impaired future reproductive function However, since this variant is paternally inherited, we can assume that this variant without the synergistic effect of other reported variants identified in our patient, cannot lead to impaired reproductive capacity, if isolated.

The CHH phenotype of our patient, consisting of micropenis, small testes and low LH, FSH and testosterone, cannot be attributed solely to the *GNRH1* p.Arg73Ter heterozygous variant, since hypogonadotropic hypogonadism 12 with or without anosmia (MIM Number 614841) is dominantly inherited. However, we may speculate that our patient’s phenotype might be due to a synergistic effect with the second CHH variant identified, the *SRA1* p.Gln32Glu. Although heterozygous variants of the *GNRH1* gene have been reported in the literature, no oligogenic case of *GNRH1* has been reported so far, whereas this is not the case for *GNRHR* [36]. Furthermore, in a recent study applying WES in a large cohort of GnRH deficient Greek patients, no *GNRH1* variant was identified either as a monogenic or as an oligogenic cause of CHH [37].

The third variant identified in our patient 1 was the *SRA1* gene variant p.Gln32Glu, which has been previously reported in compound heterozygosity in an IHH Turkish male patient in trans with the p.Ile179Thr variant [38]. It has also been identified after WES in a heterozygous 46,XY *NR5A1* patient, together with six other gene variants, supporting the concept of an oligogenic basis for DSD phenotypes [39].

The *SRA1* gene plays a dual role, encoding a long non-coding RNA and a conserved steroid receptor RNA activator protein (SRAP), which, among its other functions, regulates steroid receptor-dependent gene expression [40]. SRA1 as a steroid receptor coactivates thyroid hormone, androgen, estrogen, progesterone, glucocorticoid and retinoic acid receptors [41,42,43,44]. It has also been shown that DAX-1 functions by binding to the RNA coactivator SRA and p160 coactivator proteins and that SRA1 is important to stabilize complexes of SF-1 and DAX-1 for the recruitment of p160 coactivators in order to regulate target gene expression during steroidogenesis [45].

To date, various *SRA1* gene variants, p.Pro20Leu, p.Tyr35Asn, p.Arg126His and p.Ile179Thr, have been identified in homozygosity, compound heterozygosity, heterozygosity or as digenically inherited with *PNPLA6*, *RNF216* and *SEMA7A* gene variants in patients with CHH [38,46,47]. All but one of the patients reported with *SRA1* variants are of Turkish, Cypriot, and Greek (this report) origin, indicating the possibility of a founder effect in the East Mediterranean area.

In Patient 2, three heterozygous variants in the genes *SOX9*, *HS6ST1* and *IL17RD* were identified.

The variant p.Val95Ile of the *SOX9* gene identified in patient 2 is a novel variant. *SOX9* gene encodes the transcription factor SOX9, which is a member of the SRY-related HMG box (SOX) family of transcription factors. It is essential for both sex and skeletal development and is expressed in various tissues during embryogenesis, such as the cartilage, brain, pituitary, heart, lungs, pancreas, testes, etc. SOX proteins play an important role in the differentiation, establishment or maintenance of stem and progenitor cells of various tissues [48]. Various studies have reported the presence of potential populations of stem cells in the pituitary. In particular, it has been shown by in vitro studies that a small population of progenitor cells in the adult pituitary gland express SOX2 and SOX9 [49]. Furthermore, it was demonstrated that SOX2 and SOX9 expressing progenitor cells can self-renew and give rise to endocrine cells in vivo, suggesting that they are tissue stem cells, playing a role in the physiological maintenance of the pituitary gland as well as in the induction of pituitary tumors [50,51].

In mammals, during embryonic gonadal development, the factors SRY and SF-1 induce the expression of the *SOX9* gene. SOX9 itself then suppresses the expression of the *SRY* gene, while it takes over the differentiation of the bipotential gonad to testis. Finally, SOX9, together with SF-1, induces the differentiation of Sertoli cells in the testes [52]. The role of *SOX9* in testis formation and sex determination was first recognized in patients with campomelic dysplasia, of whom around 75% with a 46, XY male karyotype bearing one mutant *SOX* gene exhibit male-to-female sex reversal [53].

During embryonic development in mice (e13.5–14.5), the SOX9 protein is expressed in the migratory nerve cells of the brain, which are distributed in the posterior and mainly in the middle of the brain, while later on, *SOX9* is expressed in cartilage and bone [54]. Thus, the SOX9 protein has been found to be involved in the formation of craniofacial structures, and patients with gene variants present with micrognathia, cleft lip and palate, and craniofacial malformations [55].

The *HS6ST1* gene encodes the enzyme heparan sulfate 6-O-sulfotransferase 1 (HS6ST1), which participates in the biosynthesis of heparan sulfates (HSs). HSs are expressed in different developmental stages of several tissues and are involved in cell communication regulating the binding of many ligands–receptors, such as FGF8–FGFR1 [56,57]. The impaired interaction of the FGF8 ligand with the FGFR1 receptor is known to be involved in the pathogenesis of HH [58].

Pathogenic variants in the *KAL1* gene, also involved in the FGFR signaling pathway, are the main genetic cause of CHH. However, pathogenic variants of the *HS6ST1* gene are also common in patients with normosmic/anosmic IHH (approximately 2%), resembling that of KAL1-HH phenotype, but also as an oligogenic cause of CHH together with *FGFR1* and *NELF* genes [36,59,60].

The *HS6ST1* gene pathogenic variant, c.917G > A, p.Arg306Gln, was detected in patient 2 and his mother. A variant in the same codon resulting in a different amino acid, c.917G > A, p.Arg306Trp (reported as p.Arg296Trp), has been previously reported in a male patient with Kallmann syndrome and small pituitary gland [59], as was the case in our patient. The patient’s mother presented delayed puberty without anosmia, indicating an incomplete penetrance of this variant [59]. It was also shown by in vitro experiments that the activity of the p.Arg306Gln *HS6ST1* mutated enzyme is significantly reduced compared to the wild type enzyme [59].

The *IL17RD* gene (Interleukin 17 Receptor D) encodes a 739 amino acid membrane protein, member of the interleukin 17 receptor protein family, that also regulates the signaling pathway of FGF8/FGFR1, and its pathogenic variants have been associated with CHH with or without anosmia and delayed puberty and in some cases with deafness [59,61].

The *IL17RD* p.Pro566Ser identified herein has not been reported previously in CHH patients. It has been reported in a patient with evening fatigue [62], which might be due to subnormal cortisol secretion as a plausible explanation for his fatigue, although cortisol secretion has not been investigated in the patient. However, a homozygous c.1697C > T variant that results in a different amino acid change of the same codon, p.Pro566Leu, classified as probably pathogenic, has been reported in a normosmic CHH patient [33]. Two patients carrying other probably pathogenic variants of the *IL17RD* gene were identified after exome sequencing in 37 PSIS families. In these patients, the *IL17RD* variants were co-inherited with variants in other genes; specifically, one carried a *GLI2* and an *IL17RD* variant, whereas the other carried variants in three genes, *PROP1*, *IL17RD* and *SMARCA2* [56]. Although it is highly probable that the combination of several gene variants may lead to various clinical phenotypes, the contribution of each gene variant to the patient’s phenotype was not investigated in detail and, therefore, could not be established.

## 5. Conclusions

In this study, each patient was found to harbor three possible causative gene variants, after WES. Although to date numerous studies have shown that CPHD as well as PSIS have an oligogenic rather than a monogenic etiology [11,12,63,64,65], such a conclusion could not be reached in this work.

It is of interest that most of the variants detected in this study were related to CHH, indicating that there is a significant phenotypic and genetic overlap between CHH and CPHD that has also been observed in other studies [66], further supporting the notion that CPHD and CHH share common genetic background [66] and allowing us to speculate that a synergistic action of these gene variants may underlie our patient’s phenotype.

Among the six genes identified, only two (*BMP4* and *SOX9*) were expressed in the pituitary gland, and four genes (*GNRH1*, *SRA1*, *HS6ST1* and *IL17RD*), two for each patient, are known CHH genes. However, it is worth stating that two of these genes, *HS6ST1* and *IL17RD*, belong to the FGF signaling network, which plays a major role in the hypothalamic–pituitary axis. Thus far, *FGFR1* and *FGF8* gene variants have been identified to be causative genes not only for Kallmann syndrome, but also for CPHD, septo-optic dysplasia and holoprosencephaly [14,15,67]. Furthermore, the fact that the *HS6ST1* variant, p.Arg306Gln, identified in our patient has also been previously reported in a CHH patient with small pituitary gland [16] strengthens the possible role of this gene in our patient’s CPHD phenotype.

None of the identified variants herein was a de novo one. Patient 1 inherited two CHH variants, one from each parent, and a CPHD variant (BMP4), inherited from his mother. Patient 2 inherited the three variants identified from his mother, who to our knowledge does not present any of her son’s phenotypic characteristics, as is the case for patient’s 1 parents. Therefore, we can speculate that this is due either to incomplete penetrance of the gene variants, as has been previously reported for CPHD [3,11,63], or to the involvement of other etiologic factors, genetic or environmental.

Furthermore, it is highly probable that apart from the variants reported in this work, there might also exist other gene variants acting synergistically. These variants could have escaped identification either because of the WES methodological limitations, or because they are variants of genes yet unknown to be related with CPHD.

This work has several limitations, the main one being the small number of patients studied and the lack of functional studies that could possibly delineate the impact of each variant identified in the patients’ phenotypes. There are also methodological drawbacks: WES has a diagnostic turnover of 35–50%, which might be due either to technical limitations (incomplete coverage of all exons, short length reads, sequencing of GC-rich regions, repetitive sequences, and sequences with high homology), or to methodological limitations such as inability to detect intronic or structural variants, as well as the limitations of data analysis with respect to the VUS, particularly in genes with unknown function. In this study, four variants were classified as VUS; however, considering their frequency and the in silico predictions, their possible contribution to an oligogenic model of CPHD inheritance cannot be overlooked.

Further work is required to fully delineate the genetic basis of CPHD, for example, with large cohorts of patients as well as other methodologies such as whole genome sequencing, a more comprehensive sequencing option, since it provides analysis of the entire genome or the long-read WES methodology. However, the presentation of the identified gene variants in the international literature may provide further evidence to researchers for the critical role of oligogenic combinations in the pathogenesis of complex human disorders of varied penetrance and severity.

As the mystery of the genetic basis of congenital hypopituitarism is slowly unraveled, many new variants and complex combinations of variants of different genes are expected to be added to the still incomplete list of genes involved in pituitary ontogenesis and function.

## Figures and Tables

**Figure 1 cells-11-02088-f001:**
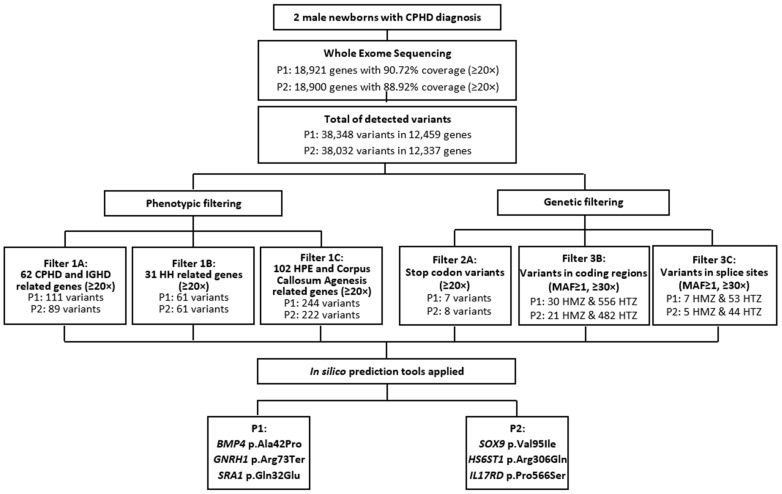
NGS analysis pipeline and filtering. Abbreviations: P1 = Patient 1; P2 = Patient 2; MAF = Minor Allele Frequency; HMZ = homozygous; HTZ = heterozygous; CPHD = Combined Pituitary Hormone Deficiency, IGHD = Isolated Growth Hormone Deficiency, HH = Hypogonadotropic Hypogonadism, HPE = Holoprosencephaly.

**Table 1 cells-11-02088-t001:** Laboratory hormone tests of the two newborns diagnosed with CPHD.

Hormones	Patients		Normal Ranges
	1	2	
TSH	5.66	<0.004	0.5–5 μUI/mL
FT4	0.784	0.776	0.8–1.8 ng/dL
ACTH	<1	3.49	7–63 pg/mL
Cortisol	0.523	2.87	6.24–18 μg/dL
D4-Andostenedione	<0.30	NA	0.05–0.45 ng/mL
DHEA-S	<15	29.4	6–21 μg/dL
LH	<0.1	<0.1	0–1.3 mIU/mL
FSH	0.123	0.226	0.1–2.4 mIU/mL
Testosterone	<20	22.5	75–400 ng/dL
GH	4.54	1.19	5–40 ng/mL
IGF-1	<15	<0.25	15–129 ng/mL
PRL	291.5	27.06	5–20 ng/mL
Aldosterone	1185	NA	50–900 pg/mL
Renin	27.6	NA	<37 ng/mL/h

Notes: TSH: Thyroid Stimulating Hormone; FT4: Free Thyroxine; ACTH: Adrenocorticotropic Hormone; DHEA-S: Dehydroepiandrosterone-sulfate; LH: Luteinizing Hormone; FSH: Follicle Stimulating Hormone; GH: Growth Hormone; IGF-1: Insulin-like Growth Factor-1; PRL: Prolactin; NA: Not Available.

**Table 2 cells-11-02088-t002:** Pathogenic variants detected by WES and their interpretation by some of the used bioinformatic tools.

Patient	Gene	Transcript ID	Nucleotide Variant	Protein Variant	dbSNP	Inheri Tance	Frequency		ACMG Classification	Prediction Tools				
							gnomAD Exomes	500 Greek Exomes		SIFT	PolyPhen2	Mutation Taster	CADD	GERP
1	*BMP4*	NM_001202.4	c.124G > C	p.[Ala42Pro;=]	rs140920120	M	0.000283	HTZ:0.002	VUS	B	LP	B	22.6	5.19
								HMZ: 0	(BP4, PS3)					
	*GNRH1*	NM_000825.3	c.217C > T	p.[Arg73Ter;=]	rs375970738	P	0.00000805	0	LP	-	-	-	14.98	4.91
									(PVS1, PM2)					
	*SRA1*	NM_001035235.4	c.94C > G	p.[Gln32Glu;=]	rs35610885	M	0.00715	HTZ: 0.008	VUS	P	LP	DC	27.6	5.01
								HMZ: 0	(BP4, PP5)					
2	*SOX9*	NM_000346.4	c.283G > A	p.[Val95Ile;=]	Novel	M	NA	0	VUS	B	LP	DC	23.3	4.24
									(PM2, BP4)					
	*HS6ST1*	NM_004807.3	c.917G > A	p.[Arg306Gln;=]	rs201307896	M	0.000706	0	P	B	LP	DC	29.9	4.78
									(PS3, PM2, PM5, PP5, PP3)					
	*IL17RD*	NM_017563.5	c.1696C > T	p.[Pro566Ser;=]	rs61742267	M	0.0144	HTZ: 0.028	VUS	B	LP	DC	22.6	5.64
								HMZ: 0	(PM5, BP4)					

Abbreviations: dbSNP = database Single Nucleotide Polymorphism, gnomAD = Genome Aggregation Database, ACMG = American College of Medical Genetics, SIFT = Sorting Intolerant From Tolerant, Polyphen2 = Polymorphism Phenotyping v2, CADD = Combined Annotation-Dependent Depletion, GERP = Genomic Evolutionary Rate Profiling, M = Maternal, P = Paternal, NA: not annotated, VUS = Variant of Uncertain Significance, B:Benign, P = Pathogenic, LP = Likely Pathogenic, DC = Disease Causing, PD = Probably Damaging, HTZ = Heterozygous, HMZ = Homozygous.

## Data Availability

Not applicable.

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
