# Peer review of "Whole Exome Sequencing Points towards a Multi-Gene Synergistic Action in the Pathogenesis of Congenital Combined Pituitary Hormone Deficiency"

_cells, 2022, doi:10.3390/cells11132088_

Round 1

Reviewer 1 Report

Sertedaki et al have performed whole exome sequencing on two patients presenting with congenital hypogonadotropic hypogonadism (CHH) and have identified heterozygous variants in multiple genes. Their conclusion is that the CHH is caused by the genetic interaction of these heterozygous variants and that the disease state is oligogenic. Such a conclusion cannot be made without functional studies. The authors recognize this as it is listed in the conclusion section as a limitation of their study, along with the limitation that whole exome sequencing does not identify all possible pathogenic variants. These are major limitations that prevent conclusions from being drawn. The identification of variants of uncertain significance is an important contribution to the knowledge base, but conclusions about the functional consequences of those variants and correlations to disease, especially when heterozygous, should not be made until verified by functional studies.

Other comments:

The methods section mentions 227 other genes that were screened after a literature search.  What are they?

The last sentences of the results section should be removed.  ”This section may be divided by subheadings. It should provide a concise and precise description of the experimental results, their inter-pretation, as well as the experimental conclusions that can be drawn.”

In the discussion section it’s the Transforming Growth Factor Beta family.  The beta is missing.

The sentence “It has also been shown that selective inhibition of bmp4 in mouse embryos results in the loss of nearly all pituitary cell lines except a few corticotrophs” should be revised.  The experiment was to ectopically express Noggin in the oral ectoderm.  Noggin is not selective for Bmp4 but also inhibits Bmp2 and 7. I might say that “Inhibition of Bmp signaling in the oral ectoderm of mouse embryos disrupts pituitary cell specification.”

There is good literature about Sox9 in pituitary stem cells, which should also be reviewed. 

Author Response

Dear Reviewer

First of all we would like to express our thanks and our appreciation for the time and effort that you have put into this review.

Sertedaki et al have performed whole exome sequencing on two patients presenting with congenital hypogonadotropic hypogonadism (CHH) and have identified heterozygous variants in multiple genes. Their conclusion is that the CHH is caused by the genetic interaction of these heterozygous variants and that the disease state is oligogenic. Such a conclusion cannot be made without functional studies. The authors recognize this as it is listed in the conclusion section as a limitation of their study, along with the limitation that whole exome sequencing does not identify all possible pathogenic variants. These are major limitations that prevent conclusions from being drawn. The identification of variants of uncertain significance is an important contribution to the knowledge base, but conclusions about the functional consequences of those variants and correlations to disease, especially when heterozygous, should not be made until verified by functional studies.

Answer: We thank you very much for your constructive comments. Indeed, we are not presenting two patients with congenital hypogonadotropic hypogonadism (CHH) but two patients with a full clinical picture of congenital hypopituitarism leading to congenital combined pituitary hormone deficiencies (CPHD). Their main clinical presentation was hypoglycemia. Concretely the first patient has been referred to our Division of Endocrinology, Diabetes and Metabolism of the First Department of Pediatrics of the National and Kapodistrian University of Athens at the Aghia Sophia Children’s Hospital from the Neonatology Department, where he has been hospitalized immediately after birth due to refractory hypoglycemic episodes with a suspicion of adrenal insufficiency. Upon clinical investigation we have noted the presence of a significant micropenis and hypoplastic scrotum with small testicles, assessed by the Prader orchidometer to have a volume of less than 1ml. The combination of hypoglycemic episodes along with the hypogonadic clinical features was reminiscent of combined pituitary hormone deficiencies and the complete endocrine and neuroimaging work-up provided the final diagnosis of combined pituitary hormone deficiencies due to congenital hypopituitarism. The second case has been referred to our First Department of Pediatrics of the National and Kapodistrian University of Athens at the Aghia Sophia Children’s Hospital for investigation by our pediatric gastroenterologists due to a significant elevation of liver enzymes with the working hypothesis of neonatal hepatitis. Upon clinical examination this patient had a significant micropenis and hypoplastic scrotum with small testicles assessed by the Prader orchidometer to have a volume of less than 1ml. During the examination the infant presented generalized tonicoclonic seizures, that were provoked by a severe hypoglycemic episode. Again, the combination of hypoglycemic episodes along with the hypogonadic clinical features was reminiscent of combined pituitary hormone deficiencies and the complete endocrine and neuroimaging work-up provided the final diagnosis of combined pituitary hormone deficiencies due to congenital hypopituitarism. The patient has been treated with substitution of hydrocortisone, thyroxine and also recombinant Growth hormone, and this treatment led to normalization of liver enzymes, highlighting again the importance of considering multiple pituitary hormone deficiencies in the context of the investigation for “congenital hepatitis”. It is therefore very important to publish such cases that presented with clinical findings reminiscent of congenital hypogonadotropic hypogonadism, due to the presence of micropenis and small testicles, there were ultimately turned out to be cases of congenital hypopituitarism, and at the molecular level that genes initially reported to be implicated in the pathogenesis of CHH can also be implicated in the pathogenesis of congenital hypopituitarism, as is the case f.ex. for genes implicated in different forms of diabetes mellitus, illustrating the overlapping molecular pathogenetic pathways

Regarding your concern that further studies, f.ex. functional studies are needed, we certainly agree with your opinion and we stated in the original manuscript that:

Lines: 707-714 Further work is required to fully delineate the genetic basis of CPHD, as for example large cohorts of patients as well as other methodologies such as Whole Genome Sequencing, a more comprehensive sequencing option, since it provides analysis of the entire genome or the long read WES methodology. However, the presentation of the identified gene variants in the international literature may provide further evidence to researchers for the critical role of oligogenic combinations in the pathogenesis of complex human disorders with various penetrance and severity.

Other comments:

The methods section mentions 227 other genes that were screened after a literature search. What are they?

Answer: The list of these genes has been added in lines 194-225.

The last sentences of the results section should be removed. ”This section may be divided by subheadings. It should provide a concise and precise description of the experimental results, their inter-pretation, as well as the experimental conclusions that can be drawn.”

Answer: The sentence was deleted.

In the discussion section it’s the Transforming Growth Factor Beta family. The beta is missing.  

Answer: It was added accordingly (line 357).

The sentence “It has also been shown that selective inhibition of bmp4 in mouse embryos results in the loss of nearly all pituitary cell lines except a few corticotrophs” should be revised. The experiment was to ectopically express Noggin in the oral ectoderm. Noggin is not selective for Bmp4 but also inhibits Bmp2 and 7. I might say that “Inhibition of Bmp signaling in the oral ectoderm of mouse embryos disrupts pituitary cell specification.”

Answer: We replaced the sentence (lines 364-365).

There is good literature about Sox9 in pituitary stem cells, which should also be reviewed.

Answer: The following has been now added in the discussion

Lines: 569-582 “Various studies have reported the presence of potential populations of stem cells in the pituitary. Particularly it has been shown by in vitro studies that a small polulation of progenitor cells in the adult pituitary gland express SOX2 and SOX9 (Fauquier T, et al., 2008). Furthermore, it was demonstrated that SOX2 and SOX9 expressing progenitor cells can self-renew and give rise to endocrine cells in vivo, suggesting that they are tissue stem cells, playing a role in the physiological maintenance of the pituitary gland as well as in the induction of pituitary tumours (Rizzoti K,et al., 2013, Andoniadou 2013).”

Reviewer 2 Report

This paper conveys interesting information that deserves publication.  Some major points to be addressed: 

Abstract: p.Arg73X is a mistake according to HGVS (X is not allowed - also refers to other nonsense variants in this paper). Use Ter or * for nonsense: p.Arg73Ter

Introduction: “3. Furthermore , various studies demonstrated …..”. Is “3” a literature position?

Patients: On physical examination – small testes. Was orchidometer used ? Please provide volume for evidencing “small testes”  

DNA isolation: What is the amount of blood used for extraction? What kind is the system used – columns with membrane? Please provide some more data

WES: What is the amount of DNA used for WES? Please provide some more data

Data filtering: filter 1 – variants with MAF <1% were excluded (I assume …?). But a causative variant in IL17RD gene (rs61742267) you provide in results,  has MAF 0,0144 (1,44%)???

Sequencing depth: Please state clearly, whether provided depth data refer to variant call or position. The same >30x is given for homozygous and heterozygous. The sequencing depth is used more frequently for position, but in such case >30x for homozygous variant means ~15x for heterozygous abnormal variant call (theoretically 15+15). If you apply only >20x, it’s possible that you will get only a few reads for abnormal variant that might be false positive (theoretically 10:10, but 15:5 or even 17:3 is possible). EMQN recommend minimal sequencing depth of 100x of germline variants in WES to avoid such situation.  For genome sequencing filtering of >30x is acceptable because DNA is fragmented and not amplified (enriched) by PCR as in WES…

Were all provided variants confirmed using Sanger? Please state it clearly in the text

Results:

Table – GenBank ID column should be entitled “Transcript ID”. In Protein Variant column the description like p.A42P/N is unacceptable. The c.124G>C in consequence leading to p.A42P not p.A42N. By the way, the rs140920120 refers to 42P/S not 42P/N… Please revise the entire column and use HGVS recommendations! Three-letter abbreviations for AA are advised. Authors mentioned that they used Varsome – this tool is providing correct variant descriptions. Please check all of them using Varsome.

VUS is a variant (not variable) of unknown significance.

At the end of results section: “This section may be divided by subheadings. ….” Looks like a lost comment of author or other reviewer … Please correct it.

Authors did not provide information regarding zygosity of identified variants in patients in results section. It is important for i.e. GNRH1 variant, since for that gene only AR mode of inheritance is recognized and reported in OMIM. This variant was further discussed in next section and there are several points to be addressed.

The sentences in discussion regarding GNRH1 variant are very confusing: “This variant has been previously reported  … () was considered as VUS since protein truncation leaves decapeptide intact (suggesting that the functionality of protein is preserved), and resulting transcript is rapidly degraded (functionality is inevitably lost...)”. Next sentence also – escape of NMD means that abnormal protein is produced (opposite to what authors wrote) and can potentially interfere with a normal protein produced from second normal allele. Potentially it can result in autosomal dominant mode of inheritance!

Hence, the provided information requires a thorough comprehension and in my opinion authors should also emphasize one important information. So far, for GNRH1 only typical, loss of function model is well evidenced and is causing AR inheritance. The impact of heterozygous variants in this gene suggesting indirectly dominant mode (via gene dosage or allele interference mechanisms) are highly speculative. Although, I agree with authors, that they also should be reported but more carefully discussed.

Another confusing sentence in conclusion section: “None of the identified variants herein was de novo one indicating either incomplete penetrance …….”. Incomplete penetrance is related to mutation type, not to its origin. Therefore usage of this combination is incorrect in my opinion. De novo variant is always expected to be pathogenic and fully penetrant.

Author Response

Dear Reviewer

First of all we would like to express our thanks and our appreciation for the time and effort that you have put into this review.

Comments and Suggestions for Authors

This paper conveys interesting information that deserves publication.  Some major points to be addressed: 

Abstract: p.Arg73X is a mistake according to HGVS (X is not allowed - also refers to other nonsense variants in this paper). Use Ter or * for nonsense: p.Arg73Ter EASY p.R73*

Answer: It has been changed to p.Arg73Ter Line 26 and throughout the text.

Introduction: “3. Furthermore , various studies demonstrated …..”. Is “3” a literature position?

Answer: We modified this literature accordingly (line 86).

Patients: On physical examination – small testes. Was orchidometer used ? Please provide volume for evidencing “small testes”  

Answer: We have now added in Line 110: “testes volume: 1ml measured by orchidometer”.

DNA isolation: What is the amount of blood used for extraction? What kind is the system used – columns with membrane? Please provide some more data

WES: What is the amount of DNA used for WES? Please provide some more data

Answer: We have revised the methods (2.2 and 2.3 sub-paragraphs) in order to add this information (lines 147-156).

Data filtering: filter 1 – variants with MAF <1% were excluded (I assume …?). But a causative variant in IL17RD gene (rs61742267) you provide in results, has MAF 0,0144 (1,44%)???

Answer: The methods have been revised accordingly (line 189).

We agree that this is a rather frequent variant. The reason why we increased the MAF value was because of the small number of identified pathogenic/likely pathogenic variants related to our patients’ phenotypes.

However, recent ClinGen recommendations suggest the decrease of weight of criterion PM2 from moderate to supporting (https://clinicalgenome.org/site/assets/files/5182/pm2_-_svi_recommendation_-_approved_sept2020.pdf) and justified as: “After substantial analysis and modeling, the SVI WG proposes this weight adjustment due to concerns that absence or rarity is given too much weight in the 2015 framework and that this type of evidence does not meet the relative odds of pathogenicity for a Moderate pathogenic evidence, estimated to be 4.33:1 (PMID:29300386). This concern is supported by findings from the Exome Aggregation Consortium (ExAC) database that 99% of identified high-quality variants have a frequency <1%, that 54% of identified high-quality ExAC variants are only seen once in the entire data set, suggesting that rarity is actually common, and that all individuals harbor variants that are absent from the rest of the population (PMID:27535533)”

Sequencing depth: Please state clearly, whether provided depth data refer to variant call or position. The same >30x is given for homozygous and heterozygous. The sequencing depth is used more frequently for position, but in such case >30x for homozygous variant means ~15x for heterozygous abnormal variant call (theoretically 15+15). If you apply only >20x, it’s possible that you will get only a few reads for abnormal variant that might be false positive (theoretically 10:10, but 15:5 or even 17:3 is possible). EMQN recommend minimal sequencing depth of 100x of germline variants in WES to avoid such situation.  For genome sequencing filtering of >30x is acceptable because DNA is fragmented and not amplified (enriched) by PCR as in WES…

Answer: The data provided referred to position reads, which has now been clarified in the text.

We totally agree with the recommendation about the number of reads. Three of the variants reported had >100x, one was 82x, one 58x and one 37x position reads. We thought that, since our variants were verified by Sanger sequencing, the small number of read 37x, is not creating a problem.

Were all provided variants confirmed using Sanger? Please state it clearly in the text

Answer: We have now added at the end of the methods section a stronger statement Lines 251-253: “All variants reported herein were confirmed by Sanger sequencing the corresponding exon of each gene in the patients and their parents.”

Results:

Table – GenBank ID column should be entitled “Transcript ID”.

Answer: We changed it accordingly in Table A2.

In Protein Variant column the description like p.A42P/N is unacceptable. The c.124G>C in consequence leading to p.A42P not p.A42N. By the way, the rs140920120 refers to 42P/S not 42P/N… Please revise the entire column and use HGVS recommendations!

Answer: With N we were referring to the normal allele. We have now changed the protein variants according to the HGVS recommendations in Table A2.

Three-letter abbreviations for AA are advised.

Answer: All variants are now expressed with three-letter abbreviations in the entire manuscript, figure A1 and table A2.

Authors mentioned that they used Varsome – this tool is providing correct variant descriptions. Please check all of them using Varsome.

Answer: For our analysis we used various tools for variant interpretation (Franklin, Varsome and InterVar). However, we noticed that Varsome changes the criteria applied quite often and this is the reason why we decided to carry out the ACMG classification by applying the criteria by ourselves. This is the classification presented in the text and Table A2.

VUS is a variant (not variable) of unknown significance.

Answer: We corrected it (line 275).

At the end of results section: “This section may be divided by subheadings. ….” Looks like a lost comment of author or other reviewer … Please correct it.

 Answer: We deleted it.

Authors did not provide information regarding zygosity of identified variants in patients in results section. It is important for i.e. GNRH1 variant, since for that gene only AR mode of inheritance is recognized and reported in OMIM. This variant was further discussed in next section and there are several points to be addressed.

Answer: We have now added Lines 455-531: “The CHH phenotype of our patient, consisting of micropenis, small testes and low LH, FSH and Testosterone, cannot be attributed solely to the GNRH1 p.Arg73Ter heterozygous variant, since Hypogonadotropic Hypogonadism 12 with or without anosmia (MIM Number 614841) is dominantly inherited. However, we may speculate that our patient’s phenotype might be due to a synergistic effect with the second CHH variant identified, the SRA1 p.Gln32Glu.

The sentences in discussion regarding GNRH1 variant are very confusing: “This variant has been previously reported  … () was considered as VUS since protein truncation leaves decapeptide intact (suggesting that the functionality of protein is preserved), and resulting transcript is rapidly degraded (functionality is inevitably lost...)”. Next sentence also – escape of NMD means that abnormal protein is produced (opposite to what authors wrote) and can potentially interfere with a normal protein produced from second normal allele. Potentially it can result in autosomal dominant mode of inheritance!

Answer: We have now added Lines 418-431: “The GNRH1 pathogenic variant p.Arg73Ter was identified in heterozygosity in our patient 1, inherited from his father. Residue R73 resides in the GAP region of the protein and results to its truncation by 19 amino acids, which although a nonsense variant it was considered as VUS, since the resulting protein truncation leaves the GnRH decapeptide intact. This variant has been previously reported in heterozygosity in a male patient investigated for absent puberty at the age of 17 6/12 years [32]. Chan and coworkers speculated that, since the premature stop codon is close to a splice junction, it may escape nonsense-mediated decay and, if this is the case, an abnormal protein is produced. Although, the effect of this variant on GNRH1 synthesis is unclear, the authors suggested that the CHH of this family with the p.Arg73Ter, might be of autosomal dominant inheritance with variable expressivity since the father, heterozygous for the variant, exhibited a mild phenotype of delayed puberty (32).”

Hence, the provided information requires a thorough comprehension and in my opinion authors should also emphasize one important information. So far, for GNRH1 only typical, loss of function model is well evidenced and is causing AR inheritance. The impact of heterozygous variants in this gene suggesting indirectly dominant mode (via gene dosage or allele interference mechanisms) are highly speculative. Although, I agree with authors, that they also should be reported but more carefully discussed.

Answer: We have now added Lines 455-536:

“The CHH phenotype of our patient, consisting of micropenis, small testes and low LH, FSH and Testosterone, cannot be attributed solely to the GNRH1 p.Arg73Ter heterozygous variant, since Hypogonadotropic Hypogonadism 12 with or without anosmia (MIM Number 614841) is dominantly inherited, we may speculate that our patient’s phenotype might be due to a synergistic effect with the second CHH variant identified, the SRA1 p.Gln32Glu. Although heterozygous variants of GNRH1 gene have been reported in the literature an oligogenic case of GNRH1 has not been reported so far, whereas this is not the case for GNRHR [37]. Furthermore, in a recent study applying WES in a large cohort of GnRH deficient Greek patients, no GNRH1 variant was identified neither as a monogenic nor as an oligogenic cause of CHH [38]. “

Another confusing sentence in conclusion section: “None of the identified variants herein was de novo one indicating either incomplete penetrance …….”. Incomplete penetrance is related to mutation type, not to its origin. Therefore usage of this combination is incorrect in my opinion. De novo variant is always expected to be pathogenic and fully penetrant.

Answer: This paragraph is now changed to:

Lines 565-573 “None of the identified variants herein, was a de novo one. Patient 1 inherited two CHH variants one from each parent, and a CPHD variant (BMP4) inherited from his mother. Patient 2 inherited the three variants identified from his mother, who to our knowledge does not present any of her son’s phenotypic characteristics, as is the case for patient’s 1 parents. Therefore, we can speculate that this is due either to incomplete penetrance of the gene variants as has been previously reported (3, 11, Dattani and Gregory 2019) or to the involvement of other aetiologic factors, genetic or environmental.”

Round 2

Reviewer 1 Report

I agree that presenting case studies that expand clinical descriptions and that present novel variants related to CPHD is important.  My main concern is the certainty with which the conclusions are presented.  "In this study, the application of WES in 2 CPHD newborns revealed the synergistic action of several gene variants mostly associated with CHH and the identification of a novel SOX9 variant, as the underlying cause of their phenotype of CPHD." Identifying these variants in the same patient does not demonstrate synergistic action.  The mother of patient 2 has the same variants and does not have CPHD.  While you use incomplete penetrance to explain this phenomenon, it could also mean that the causative variant is still unknown or that one of your variants is interacting synergistically with an unidentified variant to generate the phenotype, or many other possibilities.

Also, how do these variants interact with each other in a cell to generate a phenotype? For example, it is not apparent to me how variants in a morphogenetic protein, BMP4, a releasing hormone, GNRH, and a SRAP, SRA1, interact to modify cellular function for multiple cell types. Without functional studies the results should be presented with more care and less certainty about the presence of these alleles and a clinical phenotype.   

Author Response

Dear Reviewer

First of all we would like to express our thanks and our appreciation for the time and effort that you have put into this review.

Reviewer’s Comment

I agree that presenting case studies that expand clinical descriptions and that present novel variants related to CPHD is important.  My main concern is the certainty with which the conclusions are presented.  "In this study, the application of WES in 2 CPHD newborns revealed the synergistic action of several gene variants mostly associated with CHH and the identification of a novel SOX9 variant, as the underlying cause of their phenotype of CPHD." Identifying these variants in the same patient does not demonstrate synergistic action.  The mother of patient 2 has the same variants and does not have CPHD.  While you use incomplete penetrance to explain this phenomenon, it could also mean that the causative variant is still unknown or that one of your variants is interacting synergistically with an unidentified variant to generate the phenotype, or many other possibilities.

Also, how do these variants interact with each other in a cell to generate a phenotype? For example, it is not apparent to me how variants in a morphogenetic protein, BMP4, a releasing hormone, GNRH, and a SRAP, SRA1, interact to modify cellular function for multiple cell types. Without functional studies the results should be presented with more care and less certainty about the presence of these alleles and a clinical phenotype.

Answer

We totally understand and agree with your comments and we have performed changes both in the title but also throughout the text with respect to your really justified concerns and comments.

Concretely, we have modified the title of the manuscript into:

"Whole Exome Sequencing points towards a multi-gene synergistic action in the pathogenesis of Congenital Combined Pituitary Hormone Deficiency"

Specifically in the abstract:

Lines 24-27 “the synergistic action” is deleted and the sentence is now written as:

“In this study, the application of WES in these CPHD newborns revealed the presence of three different heterozygous gene variants in each patient. Specifically in patient 1, the variants BMP4; p.Ala42Pro, GNRH1; p.Arg73Ter and SRA1; p.Gln32Glu, and in patient 2 the SOX9; p.Val95Ile, HS6ST1; p.Arg306Gln, IL17RD; p.Pro566Ser, were identified as candidate gene variants.”

In the Discussion

The synergistic action as the underlying cause of their phenotype of CPHD has been deleted and the sentence is now written:

Line 309 : “In this study, the application of WES in 2 CPHD newborns revealed the presence of several gene variants mostly associated with CHH and the identification of a novel SOX9 variant.”

In the Conclusions

Lines 518-522   “In this study, each patient was found to harbor, three possible causative gene variants, after WES. Although to date numerous studies have shown that CPHD as well as PSIS have an oligogenic rather than a monogenic aetiology [11],[12],[63],[64],[65], such a conclusion cannot be reached in this work.”

The paragraph discussing variants inheritance has been moved further down to lines 575-583 and the following sentence has been added:

Lines: 584-587 “Furthermore, it is highly probable that there are other gene variants acting synergistically with the ones identified, which escaped identification either because of the WES methodological limitations, or because they are variants of genes yet unknown to be related with CPHD.”

We believe that after the aforementioned changes, the synergistic action of the variants initially proposed has been eliminated and it becomes clear that the variants reported in the manuscript are possibly contributing to our patients’ phenotypes (line 599).

This is also supported by the paragraph in Lines 588-600, where we discuss the methodological limitations of our work.

Reviewer 2 Report

Thank you for your corrections and explanations. I accept it.

Author Response

Dear Reviewer,

We would like to thank you for the time you have spend revising our manuscript and your valuable comments.